# Processed foods purchase profiles in urban India in 2013 and 2016: a cluster and multivariate analysis

Mehroosh Tak,[1] Cherry Law,[2] Rosemary Green,[3] Bhavani Shankar,[4] Laura Cornelsen  [5]

[1]Veterinary Epidemiology, Economics and Public Health, The Royal Veterinary College, London, UK
[2]Department of Agri-Food Economics and Marketing, School of Agriculture, Policy and Development, University of Reading, Reading, UK
[3]Centre on Climate Change and Planetary Health, London School of Hygiene & Tropical Medicine, London, UK
[4]Institute for Sustainable Food, University of Sheffield, Sheffield, UK
[5]Department of Public Health, Environments and Society, London School of Hygiene & Tropical Medicine, London, UK

**Correspondence to**
Dr Laura Cornelsen;
laura.cornelsen@lshtm.ac.uk

## ABSTRACT

**Objectives** Sales of ultraprocessed foods (UPFs) and beverages are rising in low-income and middle-income countries. Such foods are often linked with weight gain, obesity, type 2 diabetes and hypertension—diseases that are on the rise in India. This paper analysed patterns in purchases of processed and UPF by urban Indian households.

**Setting** Panel data from Kantar —Worldpanel Division, India for 2013 and 2016.

**Participants** 58 878 urban Indian households.

**Methods** We used K-mean partition clustering and multivariate regression to analyse patterns in processed food (PF) and UPF purchase for urban India.

**Results** Three-quarters of urban Indian households purchased over ten PF groups. Mean per person annual PF purchase was 150 kg. UPF purchase was low at 6.4 kg in 2016 but had grown by 6% since 2013. Cluster analysis identified three patterns of consumption, characterised by low (54% of the households in 2016), medium (36%) and high (10%) PF purchase quantities. High cluster households purchased over three times as much PFs and UPF as the low cluster households. Notably, salt purchases were persistently high across clusters in both years (>3.3 kg), while sweet snack and ready-to-eat food purchases grew consistently in all clusters between 2013 and 2016. A positive and significant association was found between household purchases of UPF and their socioeconomic status as well as ownership of durables, such as refrigerator, colour television and washing machine (all p<0.001). Spatial characteristics including size of town (p<0.05) in which the household is located were also positively associated with the purchase of UPF.

**Conclusion** Results suggest the need for tailored regional and city level interventions to curb the low but growing purchase of UPF. New data on obesity and rise of non-communicable diseases, the results are concerning given the links between lifestyle changes and the speed of urbanisation in Indian cities.

## STRENGTHS AND LIMITATIONS OF THIS STUDY

⇒ Use of a large, objective longitudinal household panel survey of processed food and ultraprocessed food (UPF) purchases for 2013 and 2016.
⇒ Representative analysis of all urban India rather than specific cities or regions.
⇒ Multivariate and cluster analysis of the patterns and associations between UPF and socioeconomic status and spatial variables.
⇒ The dataset does not include unprocessed food purchases, which would allow for a comparative analysis of dietary transitions towards UPF.
⇒ The survey data collected are for purchases and not consumption of foods.

## INTRODUCTION

As India battles the persistent double burden of malnutrition, including rising overweight or obesity rates, the prevalence of non-communicable diseases (NCDs) is posing a significant public health challenge.[1] Recent data from the National Family and Health Survey (NFHS) for 2019–2020 reveals that since 2015–2016, prevalence of obesity among children under 5 years old increased in 20 out of 22 states.[2] Overweight and obesity have also risen among adult population to 21% of women and 19% of men in 2015–2016 relative to 13% and 9.3% in 2005 and 2006, respectively.[3] NCDs have long been linked to changing dietary patterns and greater consumption of ultraprocessed foods (UPFs), in particular sweet and salty highly processed snacks and beverages.[4 5] These changes to diets reflect economic growth and rising disposable incomes for urbanising Indian households.[6] In particular, a global shift towards higher volumes of UPF and beverages purchases has been documented.[7]

While sales of UPF and beverages is stagnating in high-income countries, it is rapidly rising in middle-income countries.[8] UPF is linked with weight gain, obesity, type 2 diabetes and hypertension[9–11]—diseases that are on the rise in India. A systematic review of studies on Indian dietary patterns found an association between high intake of sweets and snacks and higher diabetes risk.[12] Thus, analyses of processed food (PF) consumption patterns can be critical to identify the entry points for interventions to prevent

diet-related diseases and for more targeted public health policy. However, detailed analyses on consumption patterns of PF and beverages and its socioeconomic determinants in India remain a significant gap in the literature.

Thus far, dietary transition analysis in India has primarily relied on the National Survey Statistics Organisation's (NSSO) Household Consumption and Expenditure Survey (HCES) that is known to not capture Indian PF consumption.[13] Additionally, the last available HCES for India is now almost a decade old (2011–2012). Using the HCES for 2011–2012, a study found that PF accounted for almost 10% of the average calorie intake in India.[14] This percentage could be as high as 30% for the richest households in urban India.[14] Since then, expenditure on packaged and PF has almost doubled per capita sales rose between 2010 and 2020 from US$26.3 to US$59.8, respectively, (at constant 2020 prices).[15] While per capita purchase quantity for PF in India remains low in comparison to other middle-income and high-income countries, there is considerable variation in dietary patterns across states.[13] Law et al[13] analysed aggregate trends in purchase of PF and found rising purchases of sweet and salty snacks in particular. However, this paper did not unpack household level determinants of PF purchases. Other existing studies rely on small regional data covering limited number of PF. For example, dietary patterns in Mumbai and Trivandrum showed high intakes of fried snacks and sweets.[16] Another study of dietary patterns among factory workers in tier 2 cities of India, Lucknow, Nagpur, Hyderabad and Bangalore, found two of the three distinct dietary patterns associated with high intake of snacks.[17] A recent study found high incidence of snack food consumption, including bakery products, savoury and sweet snacks among all age groups, gender, socioeconomic levels in the ninth largest Indian city of Pune.[18]

The aim of this paper is to analyse the patterns of processed, including ultraprocessed, food purchases in urban India in greater detail at household level. To do this. we use a panel dataset from 'Kantar—Worldpanel Division, India' on records of take-home purchases of PF and beverages in 2013 and 2016[19] from over 60 000 households on 43 237 distinct products. To understand patterns of PF and beverage purchases, we used K-partition cluster analysis and to identify socioeconomic determinants of the purchases of UPF and beverages, we conducted multivariate regression analysis. As Indian dietary patterns are influenced by regional, socio-economic and cultural preferences,[13 20] food group purchase analysis was conducted at regional level.

## METHODS
### Data
We used data from purchase records of an on-going demographically representative household expenditure panel, collected by the market insight company, 'Kantar—Worldpanel Division, India'.[19] Commercially collected data on food purchases have been increasingly used in academic research given their high frequency and high level of disaggregation. In particular, the household food purchase data collected by Kantar Worldpanel in the UK, Chile, Mexico and South Africa have been recently used to evaluate the effectiveness of food taxes and marketing regulations.[21–25] A recent systematic review concluded that commercially collected data on food purchases are a good indicator of diet at population level and particularly useful for measuring dietary patterns in countries that do not have national dietary surveys carried out regularly, such as India.[26]

This panel, which has been operating since 1981, covers 131 towns in 17 urban states in India. There was a major update after the 2011 Indian Census to ensure the panel's representativeness to the urban population with respect to the state of domicile, age of the person responsible for food purchase as well as socioeconomic status (SES). Indian households are sampled door to door and invited to participate based on these demographic characteristics. The panel is frequently reviewed by Kantar to assess the need for inviting new households and to ensure its representativeness of the 2011 Indian Census. Within each participating household, the primary shoppers are asked to record all purchases of PF taken home daily and to retain all the packaging and wrappers in preprovided containers. These diaries collect information regarding volume of purchases but not on monetary expenditure or prices. Kantar conducts regular checks over the accuracy of the purchase records by the interviewers, who compare the information in the paper diaries against packaging and wrappers retained by the households as well as existing products in pantry to avoid double counting. Purchases made for consumption outside of home are not included in these data.

Demographic and socioeconomic information for the panel of households is provided with the data for 2013 and 2016. We used purchase records from these 2 years and aggregate them to annual level to examine temporal changes in PF purchase across regions. Socioeconomic descriptors available included information on household size and composition, SES, durables owned by households (electricity in the house, ceiling fan, colour television, two wheeler, gas stove, refrigerator, washing machine, laptop or personal computer, four wheeler or air conditioner) and household residence by town size and state. Information of household composition was provided in binary variables indicating if the household includes children who are infant, under 1 year of age, between 2 and 4 years, 5 and 9 years, 10 and 14 years and 15 and 17 years of age. The SES variable was categorised as upper class (with literacy of at least 4 years and ownership of at least six durables), upper middle class (literacy of at least 4 years and ownership of five durables), middle class (literacy of at least 4 years and ownership of three durables) and lower class (illiterate with up to one durable). Towns were categorised by population size, starting from less than 100 000 people, between 100 000 and 500 000, 0.5–1 million, 1–4 million and over 4 million people.

Zones were described as East, South, West and North (zonal classification: (1) North—Delhi, Punjab, Haryana and Uttar Pradesh; (2) East—West Bengal, Bihar, Jharkhand, Guwahati (Assam) and Orissa; (3) West—Rajasthan, Maharashtra, Gujarat, Madhya Pradesh, Chhattisgarh; (4) South—Tamil Nadu, Karnataka, Kerala, Andhra Pradesh including Telangana).

We created a balanced panel of urban households to allow analysis on temporal change in PF purchases. The panel retention was high. In 2013 data, 64 941 households in urban areas reported purchases, of which 60 274 (93%) were also present in the panel in 2016. Thus, a small percentage of households discontinued participation but the attrition did not show any systematic patterns. We further excluded a small number of households (2%) due to missing information on household size. Our final dataset thus contained annual purchases from a balanced panel of 58 878 urban households.

## Food groups

The 43 237 distinct food items were grouped into 15 PF and UPF groups. PF included staples, milk, oils salt, processed wheat, tea & coffee, spices, butters & cheese and salt, while UPF included salty snacks, drinks, ready to eat foods, sweet snacks, milk drinks, frozen foods and breakfast cereals (see online supplemental table 1). UPF were defined as foods that are highly processed and contain in addition to added salt and sugar, additive such as flavours, colouring and emulsifiers, which are normally used in industrial processes only.[27]

With the exception of milk and drinks for which unit of measure is millilitres (mL), all other food groups were measured in grams (g). To aggregate the volume of purchases across food groups, we converted the volume for milk and drinks from mL to g using the conversion rate of 1 mL=1.03 g. For each year, we also created a food group diversity score (DS), which is the count of number of food groups purchased in that year, ranging from 1 to 15.

## Empirical strategy
### K-mean partition cluster analysis

As a first step, we plotted the distribution of DS across households and describe the average annual purchases of PF across SES groups. We then compared prevalence and quantity of annual purchases across the regions with a $\chi^2$ test. Second, we used K-mean partition cluster analysis to group the sampled households into clusters based on similarity of their PF purchases, allowing identification of distinct and predominant patterns in the data. Clustering was done for both 2013 and 2016 separately to analyse temporality of purchase patterns. K-mean partition uses Euclidean distances between observations to empirically estimate clusters within the dataset.[28] Partition clustering is an iterative process that minimises within-cluster variability while maximising between-cluster variability at the same time. The technique assigns observations into a predefined number of non-overlapping clusters. Each observation is assigned to the cluster with the closest

mean. New cluster means are then calculated after each observation is assigned. The process continues iteratively until no observations change clusters.[29] We chose K-means analysis as it is conceptually simple and computationally efficient. Other approaches, for example, Least Absolute Shrinkage and Selection Operator (LASSO) techniques are available but would offer meaningful advantages if dimensionality in the data were larger.[30] As the number of variables in this analysis is limited, there are therefore no apparent gains from using LASSO.

To run the cluster analysis, we calculated the quantity of foods purchased per household member in each food group by dividing household purchase quantity with household size. Clustering was then conducted for 3–8 partitions for each year separately. Once the clusters were constructed, boxplots with CIs were created for each food group by clusters to analyse purchase patterns and determine the best fitting number of clusters. Calinski and Harabasz pseudo-F index was used to identify the appropriate number of clusters, which is considered as one of the best rules to apply for this purpose.[31] It was estimated through a function of $([B](g-1))/([W](N-g))$, where B is the between-cluster sum of squares and cross-products matrix, W is the within-cluster sum of squares and cross-products matrix, g is the number of cluster groups and N stands for number of observations.[32] The larger the value of pseudo-F, the more clearly defined the cluster structures, and vice versa.

### Regression analysis

In the final step, pooled ordinary least square (OLS) regression was used to understand the socioeconomic determinants of UPF purchases. We used the logarithm of per household member purchase quantity of UPF as the outcome variable and the following explanatory variables: SES, state and town size of residence dummies, household size, binary variables describing household composition of children across ages (under 1 year, 2–4 years, 5–9 years, 10–14 years and 15–17 years) and binary variables describing durable assets owned by the household, including colour TV, refrigerator, washing machine, laptop/personal computer, four wheeler, air conditioner. We include time fixed effects in the pooled OLS model to control for macroeconomic changes over the data period. All estimations used robust SEs with clustering at state and town population level. This choice was informed by descriptive and cluster analysis where these spatial descriptors showed relevance in differentiating dietary preferences.

As the SES variable was constructed using education level and ownership of a certain number of durables, we also checked for multicollinearity with binary variables indicating ownership of durables using variance inflation factor (VIF) test. VIF for SES and durables was less than 10 suggesting no issues with multicollinearity. Additionally, we ran regression models together and separately with the SES and durable variables to check if coefficients varied. Our results were robust to these alternative specifications.

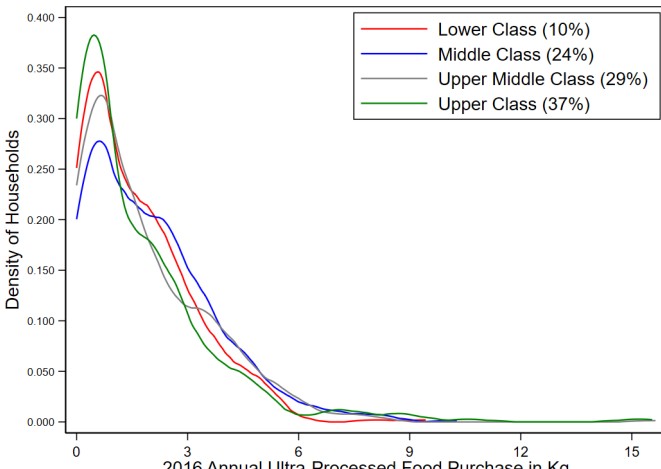

**Figure 1** Kernel density curves for ultraprocessed food (UPF) purchase by socioeconomic status in 2016.

## Patient and public involvement

There are no patients or public participation in this study.

## RESULTS

### Diversity of PF purchases

In 2016, three-quarters of the households (76%) purchased 10 food groups or more out of the 15 used in the study, implying relatively high variety of PF likely to be consumed by urban Indian households (online supplemental figure 1). Less than 1% of the households purchased all 15 food groups and less than 6% of the households bought 7 food groups or less. DS remained constant for 29% of the households, increased for 33% and declined for 37% of the households between 2013 and 2016. Most households purchased less than 10 kg of UPF per household member. Only 733 households out of 58 878 (1.3%) did not purchase any UPF in 2016 (online supplemental figure 2). The average purchase quantity of UPF in 2016 was 6.4 kg.

Figure 1 presents the kernel density curve for annual per household member UPF purchased in kilogram (kg) by SES. It shows that upper and lower class households have more probability weight at low UPF consumption levels compared with middle and upper middle class households. That is, middle and upper middle class households were more likely to purchase higher quantities of UPF than lower and higher class households. The same kernel density curve for all PF found that households with higher SES were likely to purchase higher quantities of PF (online supplemental figure 3).

### Regional variation in PF purchase

Table 1 presents the annual purchase quantities per household member by food group and zones in 2016 and the colours indicate direction of changes between 2013 and 2016. There is considerable variation in purchase patterns across zones. Overall, the purchases were highest in the North zone (218 kg annually per household member), followed by West (153 kg), East (127 kg) and South (108 kg). While UPF purchases made up only small share of these, there was an increase in the purchase of UPF overall (by 0.36 kg or ~6%, p<0.001) and in East (by 1.2 kg or ~21%, p<0001) and West (by 0.39 kg or ~9%, p<0.001) zones.

Across the food groups, most notably, the per household member purchase of ready to eat foods and sweet snacks has increased across urban India by 128 g (~22%, p<0.001) and 301 g (~15%, p<0.001), respectively. Purchase of breakfast foods including sugary cereals were very low (0.18 kg on average) but also increased considerably (~14%, p<0.001). While salt purchase declined marginally from 2013 to 2016, it remains two times as high (3.69 kg per year or 10.1 g per day) compared with recommended 5 g per day.[33] Similarly, oil purchase was high with more than 12.3 L purchased per household member per year. Purchase of drinks and milk drinks showed an overall decline by 4% and 13%, respectively, although an increase in drinks was seen in the East zone (by 0.223 L, ~17%).

Quantity of purchases per household member declined most in the North zone, driven by reduction in non-UPF purchases. Regardless, the purchase volume in most foods groups remained highest in the North zone compared with other zones. For example, drinks purchases average to 4.02 kg in the North zone, which is more than double of the urban average of 1.83 kg. Exceptions were ready-to-eat food purchases, which were much higher in the South (1.53 kg) followed by North (0.46 kg) zone. Other exceptions include milk drinks and sweet snacks that are purchased in greater quantity in the East zone (0.82 L and 3.1 kg, respectively). Per household member purchase of frozen foods was the lowest of all food groups (0.016 kg on average).

Finally, North zone also had the highest average DS in 2016 (12/15), which had increased since 2013 (p<0.001). This was followed by South and East zone (11/15). West zone had the lowest DS on average (10/15). South zone was the only zone where the decline in average DS was statistically significant (p<0.001).

### Purchase patterns: cluster analysis

We ran cluster analysis by year with number of clusters from 3–8. The Calinski and Harabasz pseudo-F index (online supplemental table 2 presents index values for each cluster by year) suggested that three clusters were the optimal partitioning for both years. This was further confirmed by visual inspection of cluster purchase patterns in box plots (due to large number of graphs, these are not presented in the paper but are available on request). Reviewing the purchasing patterns in the three partition model, we found that households fell into distinct clusters that were best characterised by purchase quantities rather than purchases of distinct food categories: low purchase, medium purchase and high purchase of PFs. These clustering patterns were consistent in 2013 and 2016 data.

**Table 1** Annual purchase quantity (g) of individual food groups per household member by zone and change from 2013 to 2016

| 2016 | Per capita consumption by food group | | | | |
|---|---|---|---|---|---|
| Zone | North | East | West | South | Total |
| Processed foods | | | | | |
| Staples | 65 054 (−364) | 39 931 (700) | 43 936 (−129) | 10 597 (496)*** | 38 663 (158) |
| Milk | 122 516 (−2780)** | 59 872 (4923)*** | 83 980 (2226)*** | 74 236 (−2637)*** | 84 689 (394) |
| Oils | 12 848 (−232)* | 12 234 (1277)*** | 14 600 (154) | 10 818 (328)*** | 12 723 (348)** |
| Salt | 3243 (−54)* | 4215 (178)*** | 3253 (−138)*** | 4158 (−114)*** | 3691 (−49)*** |
| Processed wheat | 3058 (−390)*** | 1970 (−64) | 867 (−133)*** | 824 (−431)*** | 1550 (−254)*** |
| Tea/coffee | 1052 (−86)*** | 913 (41)*** | 1082 (9) | 1026 (26)** | 1026 (0) |
| Spices | 1007 (15) | 741 (99)*** | 460 (40)*** | 593 (104)*** | 671 (65)*** |
| Butters | 241 (0) | 103 (−18)*** | 106 (16)*** | 79 (−22)*** | 127 (−5)* |
| Total PF purchases | 209 020 (−3889)** | 119 979 (7136)** | 148 284 (2045)* | 102 330 (−2251)** | 143 139 (655) |
| Ultraprocessed foods | | | | | |
| Salty snack | 1161 (−10) | 833 (184)*** | 1296 (84)*** | 623 (−94)*** | 991 (36)*** |
| Drinks | 4022 (−116) | 1334 (230)*** | 856 (−43) | 1600 (−311)*** | 1833 (−75)** |
| Ready to eat foods | 455 (56)*** | 425 (84)*** | 334 (42)*** | 1526 (315)*** | 700 (128)*** |
| Sweet snacks | 2588 (116)*** | 3099 (704)*** | 2184 (297)*** | 1713 (144)*** | 2331 (301)*** |
| Milk drinks | 105 (−28)*** | 845 (−71)*** | 91 (−18)*** | 431 (−87)*** | 339 (−50)*** |
| Breakfast cereals | 311 (63)*** | 125 (33)*** | 101 (24)*** | 199 (−19)** | 177 (22)*** |
| Frozen foods | 33 (−6)** | 12 (1) | 15 (6)*** | 6 (−5)*** | 16 (−1) |
| Total UPF purchased | 8675 (75) | 6674 (1166)*** | 4876 (390)*** | 6098 (−58) | 6387 (361)*** |
| Total PF+UPF Purchase | 217 695 (−3814)** | 126 653 (8302)*** | 153 160 (2435)** | 108 428 (−2309)** | 149 526 (1016) |
| Diversity score | 12*** | 11 | 10 | 11*** | 11*** |

Figures in parentheses show average changes between 2013 and 2016 in grams. ***p value<0.001, **p value<0.01, *p value<0.05. Cell colour: green—increase in value, red—decline in value, grey—no change. Beverages were converted from litres to kilograms by multiplying with 1.03.
PF, processed food; UPF, ultraprocessed food.

Means for key variables of interest by clusters by year are presented in table 2. From 2013 to 2016, the share of households in the low cluster declined from 64% to 54%, while the proportion of households in medium and high clusters increased from 32% to 36% and from 4% to 10%, respectively. Across the years, the quantities purchased were always smallest for low cluster, followed by medium and high cluster, suggesting that clustering patterns did not change over the 4-year period. Between 2013 and 2016, overall PF purchase volumes declined in all three clusters, while UPF purchases increased for low (p<0.01) and high purchase cluster. Purchases of PF and UPF were more than three times greater in high cluster compared with low cluster. Sweet snacks and ready-to-eat foods were two categories that had consistent increase in all three clusters while milk and milk drinks showed a consistent decrease. Although, low purchase cluster bought less of UPF, the share of UPF in their average share of food basket was higher (5.2% in 2016) than medium (3.8%) and high (3.9%) purchase clusters (we conducted analysis of variance (ANOVA) test (p value<0.001) and multivariate test of means (p value<0.001) to test statistical significance in the difference in UPF purchases by clusters for each year.

Null hypothesis of equal means was rejected confirming the difference was significant between clusters). The difference between cluster means was statistically significant (p<0.001).

The high purchase cluster households were likely to have smaller household size than low and medium purchase clusters (table 2). In line with this, households in low purchase cluster were more likely to have children in every age category compared with households in middle and high purchase clusters. In terms of geographical distribution, households in high purchase cluster were more likely to be from North zone than other zones (58% in 2013 and 49% in 2016). The medium purchase cluster included relatively similar share of households from North and West (38% and 32% respectively in 2013), which remained relatively consistent in 2016. Households from the biggest towns (>1 million in population) were more likely to be in the high purchase cluster, which may be due to a larger availability of PF in bigger towns and cities.

As expected, the medium and high purchase clusters contained a greater proportion of households from upper middle and upper class. 56% of households

**Table 2** Processed food purchase patterns for urban India for 2013 and 2016

| Purchase clusters | 2013 | | | 2016 | | |
|---|---|---|---|---|---|---|
| | Low | Medium | High | Low | Medium | High |
| Number (%) of Households | 37 331 (63.4%) | 19 059 (32.4%) | 2488 (4.2%) | 31 883 (54.1%) | 21 422 (36.4%) | 5573 (9.5%) |
| Dietary diversity | 10.7 | 11.1 | 11.1 | 10.6 | 11 | 11.2 |
| Household size | 5.1 | 4.1 | 2.7 | 5 | 4.2 | 2.9 |
| Average annual per capita purchase of food groups in g | | | | | | |
| Milk | 47 345 | 129 956 | 288 926 | 41 698 | 111 331 | 228 233 |
| Staples | 28 733 | 50 734 | 91 439 | 24 848 | 49 069 | 77 697 |
| Oils | 10 913 | 14 038 | 21 577 | 10 725 | 14 058 | 19 019 |
| Salt | 3518 | 3885 | 5953 | 3332 | 3824 | 5235 |
| Processed wheat | 1309 | 2506 | 3867 | 1060 | 1878 | 3094 |
| Tea/coffee | 843 | 1243 | 2115 | 811 | 1168 | 1709 |
| Spices | 522 | 687 | 1266 | 572 | 714 | 1069 |
| Butters & cheese | 64 | 226 | 426 | 54 | 165 | 402 |
| Total PF purchase | 93 246 | 203 276 | 415 569 | 83 100 | 182 206 | 336 458 |
| Drinks | 918 | 3248 | 6488 | 825 | 2287 | 5849 |
| Sweet snacks | 1807 | 2306 | 3266 | 2061 | 2396 | 3623 |
| Salty snacks | 713 | 1325 | 1752 | 683 | 1208 | 1925 |
| Ready to eat foods | 484 | 698 | 930 | 595 | 704 | 1285 |
| Milk drinks | 361 | 422 | 556 | 320 | 328 | 495 |
| Breakfast cereals | 86 | 246 | 494 | 85 | 228 | 513 |
| Frozen food | 5 | 33 | 69 | 5 | 21 | 55 |
| Total UPF purchase | 4375 | 8278 | 13 554 | 4573 | 7172 | 13 745 |
| % of UPF in total purchase | 4.48% | 3.91% | 3.16% | 5.22% | 3.79% | 3.92% |
| Total PF+UPF purchase | 97 621 | 211 554 | 429 123 | 87 673 | 189 378 | 350 202 |
| Percentage of households with a child in age group | | | | | | |
| Infant | 3% | 2% | 0% | 6% | 4% | 2% |
| <1 year | 5% | 3% | 1% | 3% | 2% | 1% |
| 2–4 years | 15% | 10% | 4% | 14% | 10% | 4% |
| 5–9 years | 26% | 18% | 7% | 23% | 16% | 8% |
| 10–14 years | 34% | 25% | 11% | 29% | 21% | 10% |
| 15–17 years | 26% | 19% | 8% | 23% | 18% | 9% |
| Zone | | | | | | |
| North | 11% | 38% | 58% | 9% | 33% | 49% |
| East | 26% | 12% | 7% | 26% | 14% | 12% |
| West | 31% | 32% | 21% | 30% | 34% | 26% |
| South | 32% | 18% | 14% | 34% | 20% | 13% |
| Town size | | | | | | |
| 40 lakhs+ | 26% | 25% | 18% | 27% | 23% | 23% |
| 10–40 lakhs | 31% | 44% | 53% | 30% | 43% | 47% |
| 05–10 lakhs | 15% | 11% | 7% | 15% | 11% | 9% |
| 01–05 lakhs | 14% | 11% | 14% | 15% | 11% | 11% |
| ≤01 lakhs | 14% | 10% | 8% | 14% | 10% | 10% |
| Socioeconomic status | | | | | | |

Continued

**Table 2** Continued

| Purchase clusters | 2013 | | | 2016 | | |
|---|---|---|---|---|---|---|
| | Low | Medium | High | Low | Medium | High |
| Lower class | 13% | 4% | 5% | 14% | 5% | 4% |
| Middle class | 29% | 14% | 13% | 30% | 18% | 10% |
| Upper middle class | 31% | 26% | 26% | 31% | 29% | 22% |
| Upper class | 27% | 56% | 56% | 25% | 49% | 64% |
| Durables/assets | | | | | | |
| Two wheeler | 53% | 70% | 66% | 58% | 70% | 69% |
| Refrigerator | 53% | 80% | 80% | 58% | 78% | 84% |
| Washing machine | 21% | 50% | 55% | 28% | 53% | 66% |
| Four wheeler | 19% | 38% | 36% | 23% | 35% | 40% |
| Laptop/personal computer | 10% | 24% | 29% | 16% | 27% | 38% |
| Air conditioner | 5% | 16% | 24% | 6% | 17% | 34% |

PF, processed food; UPF, ultraprocessed food.

purchasing high quantities of PF were from upper class in 2013 which increased to 64% in 2016. Furthermore, households purchasing medium and high level of all PF (including UPF) were more likely to own durables such as refrigerators, washing machine, four wheeler, laptop and air conditioner.

Figure 2 presents the share of individual PF (A) and UPF (B) food groups as percentage of total PF and UPF purchases, respectively. Figure 2A shows that the low cluster had a greater share of the foods consumed on a daily basis, such as staples, oils, tea/coffee and spices. For this cluster, almost half of the UPF purchases (45%) were sweet snacks (figure 2B). This cluster also had the highest share of ready-to-eat foods (13% of UPF) and milk drinks (7% of UPF). In comparison, the high cluster purchased a larger proportion of milk (68% of PF). They also had a higher share of drinks (43% of UPF), breakfast cereals (4% of UPF) and frozen foods (0.4% of UPF). Medium purchase cluster stood out for slightly greater share of salty snacks purchases in comparison to low and high purchase clusters.

Finally, we estimated multinomial logit models for groups identified by cluster analysis. The results are presented in online supplemental table 3). These results confirmed that higher SES, having older children and durables to be positively and significantly associated with medium and high clusters purchasing higher quantities of all PF than the low cluster.

### Determinants of UPF purchase: regression analysis

Table 3 presents the pooled OLS model for UPF quantity per household member purchased in 2013 and 2016. We found that SES, large town sizes, having children under the age 14 in the household, and ownership of durable assets were positively and significantly related to UPF purchase. Of particular importance is the SES. An upper class household purchased 34% (p<0.001) (Y=e^{0.29}=1.3364 or 34%) more UPF than a lower class household. Both middle and upper middle class households purchased more UPF than lower class household by 22% (p<0.001) and 14% (p<0.001), respectively. Households from towns with more than 4m inhabitants purchased 50% (p<0.01) more UPF than households from the smallest towns (population of less than 100 000). In comparison, households in the towns with 1–4m and 0.51m inhabitants purchased 39% (p<0.01) and 35% (p<0.05) more UPF to households from the smallest towns, respectively. An additional household member was associated with a 13% (p<0.001) less UPF purchased per member. Having children under the age of 14 years had a positive (6%–9%) and significant association (p<0.05) with UPF purchase.

Ownership of durables had a positive and statistically significant association with quantity of UPF purchased. Households that owned an air conditioner purchased 42% (p<0.001) more UPF compared with those that did not. Households who owned a computer (laptop or PC) or colour TV were associated with 20%–21% (p<0.001)

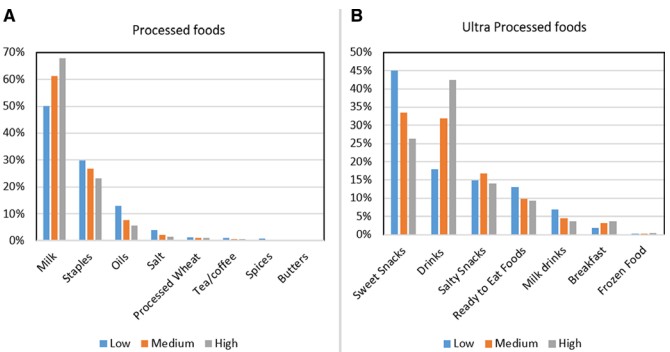

**Figure 2** Share of food groups purchase quantity by clusters in 2016.

**Table 3** Multivariate analysis of UPF purchase quantity (g) per household member (UPF qphm)

| Outcome variable: log(UPF qphm) // independent variables | Coefficient | SE | P value | 95% CI | |
|---|---|---|---|---|---|
| Base—SES—lower class | | | | | |
| SES—middle class | 0.127 | 0.032 | <0.001 | 0.062 | 0.192 |
| SES—upper middle class | 0.196 | 0.043 | <0.001 | 0.109 | 0.282 |
| SES—upper class | 0.290 | 0.049 | <0.001 | 0.193 | 0.387 |
| Base—town population<100k | | | | | |
| Town population—500k≥X>100k | 0.154 | 0.133 | 0.252 | −0.112 | 0.421 |
| Town population—1mil≥X>500k | 0.300 | 0.129 | 0.023 | 0.042 | 0.557 |
| Town population—4mil≥X>1mil | 0.331 | 0.119 | 0.007 | 0.094 | 0.568 |
| Town population—>4mil | 0.402 | 0.143 | 0.007 | 0.117 | 0.688 |
| Household size | −0.143 | 0.005 | <0.001 | −0.154 | −0.132 |
| Infant | 0.067 | 0.028 | 0.020 | 0.011 | 0.122 |
| Children under 1 year | 0.080 | 0.027 | 0.004 | 0.027 | 0.133 |
| Children 2–4 years | 0.086 | 0.011 | <0.001 | 0.065 | 0.108 |
| Children 5–9 years | 0.088 | 0.013 | <0.001 | 0.062 | 0.115 |
| Children 10–14 years | 0.058 | 0.010 | <0.001 | 0.038 | 0.077 |
| Children 15–17 years | 0.008 | 0.011 | 0.430 | −0.013 | 0.029 |
| Durable: colour TV | 0.183 | 0.040 | <0.001 | 0.103 | 0.264 |
| Durable: refrigerator | 0.090 | 0.024 | <0.001 | 0.043 | 0.138 |
| Durable: washing machine | 0.124 | 0.031 | <0.001 | 0.062 | 0.186 |
| Durable: laptop/PC | 0.194 | 0.025 | <0.001 | 0.145 | 0.243 |
| Durable: four wheeler | 0.164 | 0.037 | <0.001 | 0.089 | 0.239 |
| Durable: air conditioner | 0.352 | 0.045 | <0.001 | 0.261 | 0.443 |
| Time effect (base 2013) | 0.016 | 0.027 | 0.563 | −0.039 | 0.070 |
| Constant | 1.422 | 0.152 | <0.001 | 1.119 | 1.725 |
| Observations | 58878 | | | | |
| State effect | Yes | | | | |

Pooled ordinary least square with robust SEs clustered at state and town population level; ***p<0.001, **p<0.01, *p<0.05.
SES, socioeconomic status; UPF, ultraprocessed food.

more UPF purchase than those households that did not own them (note that only 22% of households owned laptop/PCs but 98% owned a colour TV in 2016). Ownership of a four wheeler and washing machine increased purchase by 18% (p<0.001) and 13% (p<0.001) respectively, while ownership of a refrigerator only increased purchase by 9%. About a third (37%) of the households did not own a refrigerator in 2016 suggesting that households might be primarily purchasing UPF that do not require cold storage.

## DISCUSSION

This paper used a unique dataset for urban Indian household purchases to assess patterns and socio-economic determinants of processed, and UPF purchases in 2013 and 2016. We found that three-quarters of urban Indian households purchase a higher variety of PF (10 out of 15 food groups). However, 60% of them purchased PF at small quantities that was less than 150kg per household member annually of which vast majority were commonly consumed products such as milk, oil, atta, rice and salt.

This analysis, to our knowledge, is the first to use household purchase data to examine purchases of UPF in urban India and we found that the average annual purchase of these foods was relatively low at 6.4 kg per household member, but importantly it had increased by 6% between 2013 and 2016. In comparison, for example, in the USA, share of calories from UPF consumption increased by 7.5% between a 16 year time period from 2002 and 2018.[34] We also found significant regional variation in UPF purchase, as North Indian households purchase on average 8.7 kg UPF per household member annually. This was followed by East (6.7 kg), South (6.1 kg) and West (4.9 kg) zones. However, increases in UPF purchases were most notable in East (21%) and West (9%). Northern households also purchased higher quantity of all PFs. On average, they purchased 217 kg of PF per household member in comparison to the national average of 150 kg. Southern

households purchased less than half of northern average (108 kg), even though the region has high levels of urbanisation. Higher urbanisation levels in South India may not translate to high consumption of PF and UPF because of the regional food cultures and regional heterogeneity in preferences in food consumption. Our analysis confirms that regionality determines purchase patterns of PF and UPF, as suggested by previous dietary studies on India.[20 35] A clear health risk is the high average level of annual salt purchases that was at over 3.7 kg per person, which is nearly two times as high as WHO guideline of 1.8 kg per year.[36] Both quantity and preferences for food groups varied across urban regions in the country.

Cluster analysis found three distinct purchase clusters among urban Indian consumers—low (54% of households in 2016), medium (36% of households) and high (10% of households) purchase clusters in 2016. Despite the short period of time between two data points, we saw a shift towards medium and high purchase clusters in 2016 with greater share of households accounted by these two clusters compared with 2013. Medium and high purchase clusters were more likely to have households from higher SES and living in big cities. Households in high purchase cluster bought more of every type of food groups, including three times more of UPF than low cluster. However the low purchase cluster had the greatest volume share of more commonly consumed PF such as rice, atta, oil, tea, coffee and spices. High purchase cluster had a higher share of higher value PF such as milk, butter & cheese, drinks, breakfast cereals and frozen foods. Overall purchases of UPF were relatively low, even in the high purchase cluster (eg, 5.7 L of drinks, 3.6 kg of sweet snacks and 1.9 kg of salty snacks per year) but it was increasing both in high and low purchase clusters. Sweet snacks and ready-to-eat foods in particular showed the greatest increase between 2013 and 2016 with both being the two most prominent UPF purchased by low purchase cluster. Finally, we found that quantity of UPF purchased was positively associated with SES, town size and presence of children younger than 14 years as well as ownership of durables, such as refrigerator and a four wheeler vehicle.

Our results are consistent with the limited literature available in this area. For example, Baker *et al*[7] who used Euromonitor International Passport database to estimate sales of UPF and beverages found that in 2019, per capita sales of PF stood at less than 50 kg in India, while the compounding annual growth rate for sales between 2009 and 2019 was 6%. Regional differences for dietary patterns were also observed by other studies of Indian diets.[13 20 35] For example, using NSSO food purchase data for 2011–2012, Western Indian households were found to have lower dietary DS than the rest of the country.[20] Our analysis also found diversity of PF purchased to be one group lower for households in Western zone, on average. A recent study based on 24-hour recall data for North and South Indian households found salt intake to be 11 g per day or 4 kg annually, similarly to our findings.[37] The study also found that salt intake was mainly from added salt during cooking. Our results confirm that in comparison to Latin American countries, purchase of sugar-sweetened beverages in India is relatively low.[38]

The positive association between PF consumption and socioeconomic position, however, has been observed in studies conducted in other developing countries.[39–43] In Mexico, for example, high SES individuals were found to have 3.4%–7.8% point greater share of energy contribution of UPF in the diet and the study also found regional differences in UPF consumption. In Brazil, similarly, the contribution of UPF to energy in the diet has been found to be 20% less among the lowest income group compare to the highest. To the contrary, the evidence in developed countries (eg, the USA,[44] Canada,[39] Australia,[45] Portugal[46]) seems to indicate that UPF consumption is higher among lower SES individuals or households. Moubarac *et al* hypothesise that this is due to UPF products costing relatively more in developing countries,[39] whereas Magalhães *et al* point that as countries become wealthier its growing middle classes may be consuming more UPF to exhibit SES.[39 46] While we were able to provide new insight into dietary behaviours in urban India over time and by population groups combining cluster analysis with multivariate analysis to investigate determinants of UPF purchases, our work has limitations. First, the dataset did not include unprocessed food purchases, which would allow a more refined analysis of dietary transitions towards processed and UPF. Despite this, the analysis of processed and UPF purchases is relevant on its own, particularly as the NSSO has thus far lacked detail on these foods. In future, should new NSSO data still lack detail on PF a useful avenue for research could be a matching analysis of NSSO and Kantar data. Law *et al*[13] found that for most food items, which could be compared across the two datasets, such cold beverages, milk (in liquid form) and edible oils, the difference between NSSO data and Kantar were small.[13]

Second, also inevitable due to lack of data, is the missing information of out-of-home purchases which include purchases made on the go as well as food consumed in restaurants, cafes or work and study places. Additionally, we do not know the exact composition of the household in terms of age and gender and therefore have to assume that food and beverage purchases are shared equally across household members. Finally, Kantar data is based on purchase of food items rather than actual consumption. These limitations, however, are not unique to these data. Research using similar granular purchase data in high-income countries also make this assumption.[47] Regardless of the limitations, our analysis provides new insight into diets in India and helps to improve existing evidence on the nature of processed and UPF in urban Indian diets, which are often linked to increasing obesity and diet-related NCDs.

Our findings underline important differences and changes in dietary patterns over time in urban Indian population. The results have critical implications for ongoing debates on the role of processed and UPF in low-income and middle-income countries. Key concerns are

low but rising purchase of UPF, persistently high levels of salt purchase and growing trends towards sweet and salty snacks, breakfast cereals and ready-to-eat foods across the country but particularly in Northern India. Significant role of SES, town size and regional preferences suggests the need for tailored regional and city level interventions to curb the low but growing purchase of UPF. The results are concerning given the links between lifestyle changes and speed of urbanisation[48] especially in tier 2 and 3 cities of the country along with the recently released NFHS survey results for 2019–2020 that found a dramatic rise in obesity among children under 5 in 20 out of 22 Indian states.[49]

**Contributors** MT, LC and CL designed the study. MT conducted the data analysis. MT, LC, CL, RG and BS interpreted the data and the analysis findings. MT wrote the first draft. CL, LC, BS and RG critically revised and edited the manuscript for intellectual content. LC was the overall guarantor. All authors have read and approved the final draft.

**Funding** This study forms part of the Sustainable and Healthy Food Systems (SHEFS) programme supported by the Wellcome Trust's Our Planet, Our Health programme [grant number: 205200/Z/16/Z]. LC is funded by a UK Medical Research Council fellowship (grant number MR/P021999/1).

**Competing interests** None declared.

**Patient and public involvement** Patients and/or the public were not involved in the design, or conduct, or reporting, or dissemination plans of this research.

**Patient consent for publication** Not applicable.

**Provenance and peer review** Not commissioned; externally peer reviewed.

**Data availability statement** Data may be obtained from a third party and are not publicly available. Purchase data were provided by "Kantar-Worldpanel Division, India". The terms of our data agreement with Kantar mean that we cannot share these data

**ORCID iD**
Laura Cornelsen http://orcid.org/0000-0003-3769-8740

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
