## [Reviewer comments · BMJ Open]

ARTICLE DETAILS

TITLE (PROVISIONAL)	Processed foods purchase profiles in urban India in 2013 and 2016: a cluster and multivariate analysis
AUTHORS	Tak, Mehroosh; Law, Cherry; Green, Rosemary; Shankar, Bhavani; Cornelsen, Laura

VERSION 1 – REVIEW

REVIEWER	Rahman , Andaleeb Cornell University
REVIEW RETURNED	30-Apr-2022

GENERAL COMMENTS	Using longitudinal data (2013 and 2016) from Kantar – Worldpanel Division, this paper, analyses spatial patterns in the rising purchase of ultra-processed foods and beverages (UPF) in urban India through using K-mean partition clustering and multivariate regression methods. Dietary changes in LMICs have led to the rising global burden of non-communicable diseases (NCDs) often mediated through greater consumption of UPF. Yet, there is sparse research to characterize such dietary changes and the role of UPFs which makes this paper an important contribution to the nexus of food and health. Here are my comments on the paper: • On the dataa. The authors need to convince more on the usefulness of this dataset and on the sampling methodology adopted to be a representative sample at the national and subnational level. It would be good to cite academic papers or peer-reviewed research reports which use the same for greater credibility. A small section, preferably in the appendix, on the sampling methodology adopted would be great too.b. I understand that the data reports purchases and not actual consumption. These purchases could be for daily household use or for events or public gathering. In translating from purchase to consumption, there could be distributional issues. For example, spatial inequality across urban centers might affect household consumption patterns. I believe this may not be huge concern but it would be nice if author(s) can find a way to compare this against NSSO surveys, which are the benchmark consumption surveys in India. Recently some papers have also used CMIE consumption data but I would suggest comparing against those only if the authors have access to the same as it is a propriety dataset.• On the methodsa. It is important to explain why is the K-mean partitioning for socioeconomic determinants more useful here than other
--

	methods? Predictive machine learning predictive algorithms such as LASSO might do a similar job of purchase patterns. Similarly, quantile regression models might mimic the clusters b. While not necessary to employ that, I would suggest author(s) to discuss between the choice of their chosen methods a bit more convincingly.  • On the results and conclusions  a. The consumption of processed and ultra-processed food in South India which is more urbanized and hence expected to have greater consumption of the same, is on the contrary lowest. How would the author(s) explain this finding? Also, focusing on the change between such a short period of time 2013-16 may not be very informative as diet or behavioral changes are sticky. b. It would be more useful for the reader if the results are interpreted in the line of existing scholarship (for India or other developing countries) on how diet purchased food or UPF consumption changes with socioeconomic indicators. c. The manuscript would be more useful if the 'what is new and interesting' aspect of this research, despite the data limitations, is reflected throughout while discussing results. • On Style and copyediting  a. There some conflict issues which leads to missing references, for ex. pp.7, line 41. Please check for many similar editorial mistakes.
--	--

REVIEWER	Ahmed, Asma Aga Khan University, Department of Medicine
REVIEW RETURNED	21-Jun-2022

GENERAL COMMENTS	Highlights a significant issue and should be considered for publication.
--

VERSION 1 – AUTHOR RESPONSE

2	 • On the data  a. The authors need to convince more on the usefulness of this dataset and on the sampling methodology adopted to be a representative sample at the national and subnational level. It would be good to cite academic papers or peer-reviewed research reports which use the same for greater credibility. A small section, preferably in the appendix, on the sampling methodology adopted would be great too. 	Kantar Worldpanel data has been used recently in many studies investigating purchases and changes in purchases due to food policies such as taxes on sugary drinks or labelling of unhealthy foods (Aguilar, Gutierrez, & Seira, 2021; Smith, Cornelsen, Quirmbach, Jebb, & Marteau, 2018; Stacey et al., 2021; Taillie et al., 2021; Yau et al., 2022). Bandy, Adhikari, Jebb, and Rayner (2019) have systematically reviewed studies used commercial food purchase data, including from Kantar, and concluded that they are generally considered a good indicator of diets at population level and particularly useful for countries which do not have national dietary survey carried out regularly. In India, Kantar samples urban households door-to-door based on their socio-economic status, age of the person responsible for food purchase as well as the state of domicile. This panel has been operating since 1981, covering 131 towns in 17 urban states in India. Kantar
---	--	---

		regularly reviews the panel to assess the need for recruiting new households and to ensure its representativeness of the 2011 Indian Census. We have now included these additional details in section 2.1 of the manuscript.
3	b. I understand that the data reports purchases and not actual consumption. These purchases could be for daily household use or for events or public gathering. In translating from purchase to consumption, there could be distributional issues. For example, spatial inequality across urban centers might affect household consumption patterns. I believe this may not be huge concern but it would be nice if author(s) can find a way to compare this against NSSO surveys, which are the benchmark consumption surveys in India. Recently some papers have also used CMIE consumption data but I would suggest comparing against those only if the authors have access to the same as it is a propriety dataset.	We have done this in a previous paper. Law et al (2019) compared 2013 data from Kantar to NSSO 2011-12 data.. The analysis found that for most food items that could be compared across the two data sources such cold beverages, milk (in liquid form) and edible oils, the difference between NSSO data and Kantar were small. The exception was butter, for which Kantar estimates were notably higher than NSSO. Law et al note that the discrepancy could be due to: “i) an increase in consumption of these foods in 2013; ii) an underestimation from NSS data as it only covers 30 days and purchase of dairy products may not take place sufficiently frequently; iii) an overestimation bias in the data from ‘Kantar – Worldpanel Division, India’. The latter is relatively less likely as Kantar checks purchase records against the wrappers of the purchases regularly by interviewers.” Unfortunately, we do not have access to CMIE data and thus cannot provide a similar analysis. However, the comparison of Kantar data to NSSO by Law et al (2019), with whom we share co-authors, suggests that Kantar data compares well to the benchmark consumption survey. We have added that NSSO and Kantar data are comparable as suggested by Law et al (2019) in the manuscript on page 14.
4	On the methods a. It is important to explain why is the K-mean partitioning for socioeconomic determinants more useful here than other methods? Predictive machine learning predictive algorithms such as LASSO might do a similar job of purchase patterns. Similarly, quantile regression models might mimic the clusters. b. While not necessary to employ that, I would suggest author(s) to discuss between the choice of their chosen	We thank the reviewer for raising this point. A discussion on methods available to conduct dietary pattern analysis has been now included in the manuscript. K-means is conceptually simple and computationally efficient, but LASSO is advantageous as an alternative under high dimensionality (when the number of variables becomes large). In this case, the number of variables is limited, and there are no apparent gains from using LASSO or other alternatives (Narayanan, Babu, & Kaimal, 2015). Other popular methods include Principal component analysis (PCA). However, K-means method provides algorithm based clustering to mutually exclusive categories maximising the distance between clusters rather than relying on researcher to classify observations into groups. With respect to quantile regression models, we are of the opinion that these are not as useful as the clustering technique due to its limited ability to consider all food groups together. This could possibly be done using interactions in the model but clustering allows to consider the full basket in a more straightforward manner.

	methods a bit more convincingly.	A discussion on choice of methods has been included in section 2.3 of the manuscript.
5	On the results and conclusions a. The consumption of processed and ultra-processed food in South India which is more urbanized and hence expected to have greater consumption of the same, is on the contrary lowest. How would the author(s) explain this finding? Also, focusing on the change between such a short period of time 2013-16 may not be very informative as diet or behavioral changes are sticky.	The reviewer raises two critical points. Existing studies analysing NSSO data find differences in regional food consumption patterns (Tak et al, 2019; Chaudhary et al, 2020 among others). The authors think higher urbanisation levels in South India may not translate to high consumption of PF and UPF because of the regional food cultures and regional heterogeneity in preferences in food consumption. In our analysis, we also find regionality to determine purchase patterns. For example, South Indian households purchased the highest amount of ready to eat foods and lowest amount of sweet snacks. We acknowledge that regional food preferences do not fully explain the low levels of PF and UPF purchase in the South. We have included a discussion on page 13 of the manuscript. Due to the limited number of observations for rural households in the Kantar dataset, we were unable to assess the association with urbanisation and rurality. As such this is out of scope of this study. The reviewer's point on dietary habits being sticky is well-taken, and well-documented. We acknowledge that the temporal window for discussing change is shorter than ideal, and as such this as a shortcoming of our paper. However, our analysis show that households move between clusters as more households were part of the high cluster in 2016 compared to 2013. Thus temporal analysis even for short periods may allow for a nuanced understanding on purchase/ consumption patterns, especially for the case of processed foods, a very dynamic sector involving considerable effort by the food industry to boost consumption.
6	b. It would be more useful for the reader if the results are interpreted in the line of existing scholarship (for India or other developing countries) on how diet purchased food or UPF consumption changes with socioeconomic indicators.	Thank you for this suggestion. Given the limited space and vast literature on socio-economic indicators and diet or food purchasing we have focused on studies regarding UPF consumption and differences between developing and developed countries when it comes to socio-economic disparities. We have added a paragraph on this in the discussion.
7	c. The manuscript would be more useful if the 'what is new and interesting' aspect of this research, despite the data limitations, is reflected throughout while discussing results.	Thank you for this suggestion. We have edited the discussion to reflect this.
8	 • On Style and copyediting a. There some conflict issues which leads to missing references, for ex. pp.7, line 41.	Thank you for spotting the typos. These have been fixed.

Please check for many similar editorial mistakes.	
---	--

Bibliography

- Aguilar, A., Gutierrez, E., & Seira, E. (2021). The effectiveness of sin food taxes: Evidence from Mexico. *J Health Econ*, 77, 102455. doi:10.1016/j.jhealeco.2021.102455
- Bandy, L., Adhikari, V., Jebb, S., & Rayner, M. (2019). The use of commercial food purchase data for public health nutrition research: A systematic review. *PLOS ONE*, 14(1), e0210192. doi:10.1371/journal.pone.0210192
- Narayanan, L., Babu, A. S., & Kaimal, M. R. (2015). *Projected Clustering with LASSO for High Dimensional Data Analysis*, Cham.
- Smith, R. D., Cornelsen, L., Quirnbach, D., Jebb, S. A., & Marteau, T. M. (2018). Are sweet snacks more sensitive to price increases than sugar-sweetened beverages: analysis of British food purchase data. 8(4), e019788. doi:10.1136/bmjopen-2017-019788 %J *BMJ Open*
- Stacey, N., Edeka, I., Hofman, K., Swart, E. C., Popkin, B., & Ng, S. W. (2021). Changes in beverage purchases following the announcement and implementation of South Africa's Health Promotion Levy: an observational study. *Lancet Planet Health*, 5(4), e200-e208. doi:10.1016/s2542-5196(20)30304-1
- Taillie, L. S., Bercholz, M., Popkin, B., Reyes, M., Colchero, M. A., & Corvalán, C. (2021). Changes in food purchases after the Chilean policies on food labelling, marketing, and sales in schools: a before and after study. *Lancet Planet Health*, 5(8), e526-e533. doi:10.1016/s2542-5196(21)00172-8
- Yau, A., Berger, N., Law, C., Cornelsen, L., Greener, R., Adams, J., . . . Cummins, S. (2022). Changes in household food and drink purchases following restrictions on the advertisement of high fat, salt, and sugar products across the Transport for London network: A controlled interrupted time series analysis. *Plos Medicine*, 19(2), e1003915. doi:10.1371/journal.pmed.1003915

1

VERSION 2 – REVIEW

REVIEWER	Rahman , Andaleeb Cornell University
REVIEW RETURNED	23-Aug-2022
GENERAL COMMENTS	I would like to thank the authors for a satisfactory revision of the manuscript.